# Preclinical Safety Profile of an Oral Naringenin/Hesperidin Dosage Form by In Vivo Toxicological Tests

**Carla Georgina Cicero-Sarmiento** [1,2], **Rolffy Ortiz-Andrade** [1,*], **Jesús Alfredo Araujo-León** [3], **Maira Rubí Segura-Campos** [4], **Priscila Vazquez-Garcia** [1,3], **Héctor Rubio-Zapata** [5], **Efrén Hernández-Baltazar** [6], **Victor Yañez-Pérez** [1], **Amanda Sánchez-Recillas** [1], **Juan Carlos Sánchez-Salgado** [7], **Emanuel Hernández-Núñez** [8] and **Durcy Ruiz-Ciau** [3]

1 Laboratorio de Farmacología, Facultad de Química, Universidad Autónoma de Yucatán, Merida 97069, Mexico; cicero_carla@comunidad.unam.mx (C.G.C.-S.); priscila.vgarcia@hotmail.com (P.V.-G.); viyaperez@gmail.com (V.Y.-P.); amanda.sanchezrecillas@gmail.com (A.S.-R.)

2 Programa de Maestría y Doctorado en Ciencias Médicas, Odontológicas y de la Salud, Facultad de Medicina, Universidad Nacional Autónoma de México, Ciudad Universitaria, Mexico City 04510, Mexico

3 Laboratorio de Cromatografía, Facultad de Química, Universidad Autónoma de Yucatán, Merida 97069, Mexico; jalfredoaraujo@gmail.com (J.A.A.-L.); rciau@correo.uady.mx (D.R.-C.)

4 Facultad de Ingeniería Química, Universidad Autónoma de Yucatán, Merida 97203, Mexico; maira.segura@correo.uady.mx

5 Facultad de Medicina, Universidad Autónoma de Yucatán, Merida 97000, Mexico; hector.rubio@correo.uady.mx

6 Laboratorio de Tecnología Farmacéutica, Facultad de Farmacia, Universidad Autónoma del Estado de Morelos, Cuernavaca 62209, Mexico; efrenhb@uaem.mx

7 Hypermedic MX, Mexico City 04930, Mexico; juanc.sanchez@live.com

8 Departamento de Recursos del Mar, Centro de Investigación y de Estudios Avanzados del Instituto Politécnico Nacional–Unidad Mérida, Merida 97205, Mexico; emanuel.hernandez@cinvestav.mx

* Correspondence: rolffy@correo.uady.mx; Tel.: +52-999-922-570

**Abstract:** We developed a naringenin–hesperidin molar mixture (MIX–160) with proven antihyperglycemic and vasorelaxant activity in preclinical studies. A solid dosage form was manufactured to improve the bioavailability properties. In the current study, we sought to evaluate the oral preclinical toxicity of the MIX–160 dosage form, which showed no mortality or significant changes in the body weight, food consumption and tissue/organ mass in rats. Three daily oral doses (50, 300 and 2000 mg/kg of MIX–160) were assayed for 28 days. The results showed no structural abnormalities in the histological analysis and no significant changes ($p > 0.05$) in the liver biochemical markers (total bilirubin, AST and ALT) compared to the control group. The above findings showed that the MIX–160 dosage form did not exhibit relevant toxic effects, which suggests its potential safety as a drug candidate for clinical studies.

**Keywords:** acute oral toxicity; flavonoids; hesperidin; naringenin; 28–day oral toxicity

## 1. Introduction

Flavonoids are naturally occurring compounds with several biological effects, such as antioxidant [1,2], antineoplasic [3], anti–inflammatory [4], antihyperglucemic [5,6] and antihypertensive [7]. These natural products have been identified in different fruits and vegetables, such as tomato, mandarin, grapefruit, lemon and orange [8,9]. Particularly, six major flavonoids are widely distributed in sweet orange species: rutin, quercetin, hesperetin, naringin, herperidin and naringenin [10,11].

Early preclinical studies have been conducted where their biological effects are demonstrated. Thus, intragastric administration of naringenin (50 mg/kg) induced a significant decrease in plasma glucose in non–insulin–dependent diabetic rats [12]. The oral administration of hesperidin (200 mg/kg) demonstrated a hypolipidemic effect in cardiotoxicity-induced rats [13].

In this context, our research group developed a mixture of naringenin–hesperidin, MIX–160. This preparation demonstrated improved absorption when administered as a mixture (161 mg/kg) compared with a single dose of naringenin (92 mg/kg) and hesperidin (69 mg/kg) [14]. Oral administration of MIX–160 produced a significant decrease in systolic and diastolic blood pressure in spontaneously hypertensive rats at 5 and 7 h post-administration [15]. Furthermore, subchronic oral administration of the mixture for 30 days improved carbachol–induced relaxation and exerted less vascular contractibility in norepinephrine–induced contraction [15].

Currently, there are drug products based on flavonoid–containing formulations. For instance, Fabroven® is an approved medicine indicated for chronic venous insufficiency, which consists of a combination of *Ruscus aculeatus* L., root extract, hesperidin methyl chalcone and ascorbic acid. In the other hand, Daflon® is another approved medicine that contains 90% micronized diosmin and 10% hesperidin for the same indication [16,17]. Currently, flavonoids are considered to be potential drug molecules due the multiple beneficial effects on cardiovascular and metabolic disorders. Thus, it is imperative to promote the manufacturing of innovative flavonoid–containing drug products for highly prevalent cardiometabolic diseases, such as hypertension and diabetes, as well as associated complications.

Due its combined antihypertensive and antidiabetic properties of naringenin and hesperidin confirmed in animal models, MIX–160 could be a potential alternative as a therapeutic agent. However, it is necessary to assess its short– and long–erm toxicity. Based on Sigma Aldrich specifications, naringenin and hesperidin are classified as category 4 acute toxicity substances. This indicates a median lethal dose ($LD_{50}$) between 300 and <2000 mg/kg [18]. Single and sub–chronic toxicity tests of a herbal formula containing hesperidin at the dose of 2000 mg/kg revealed no mortality and no abnormal signs in the body weight or histologic tests. In this research, the $LD_{50}$ was more than 2000 mg/kg [19].

A subchronic toxicity study of methyl hesperidin in B6C3F1 mice at a dose of 5.0% in the diet for 13 weeks revealed no obvious toxic effects in mice of either sex [20]. Peibo Li et al. proved that naringin was practically non–toxic for Sprague–Dawley rats in an oral acute toxicity study and the no–observed–adverse–effect–level (NOAEL) of naringin in rats was greater than 1250 mg/kg/day when administered orally for 13 consecutive weeks [21]. Other results reported by Andrade-Ortiz et al. were also consistent with these data, thus, positioning these flavonoids as low–risk, useful substances for drug development [22].

We aimed to evaluate the acute and sub–chronic oral toxicity of MIX–160 by histological analysis, body weight changes and biochemical marker quantification. Assessments were made at three different doses (50, 300 and 2000 mg/kg) as a single dose administration and after repeated doses of 161 mg/kg for 28 days.

## 2. Materials and Methods

### 2.1. Chemicals and Drugs

Pharmaceutical naringenin and hesperidin (>90%) for tablet manufacturing were purchased from Sigma Aldrich Co. (St. Louis, MO, USA) with microcrystalline cellulose AVICEL® PH 102 DUPOINT from Dasan, (Mexico City, Mexico), agglomerated α–lactose monohydrate Tablettose® 70 from Meggle AG (Wasserburg am Inn, Germany), sodium croscarmellose from JRS Pharma (Patterson, NY, USA); sodium lauryl sulfate from Cosmopolita Drug Store (Mexico City, Mexico), and magnesium stearate from Central de Drogas, SA de CV (Mexico City, Mexico).

### 2.2. Technopharmaceutical Development of a Solid Dosage Form

For tablet manufacturing, all excipients were selected based on the literature [23] and considering the physicochemical properties of the flavonoid powders. Four different excipient–active substance mixtures were made and submitted to 50 °C and 40% humidity conditions in an incubation oven for 21 days. The mixtures were analyzed using the infrared spectrum (IR) to identify any degradation signals. Excipients that showed degradation

under the described conditions were eliminated. A "22 + 1 central point design" was used for selecting the optimum surfactant and disintegrant concentrations. Five pilot batches were used by direct compression in a 10 mm pricker (1400 psi) using a hydraulic press. Pilot batches comprised 25 tablets each. Ten tablets were for the pharmacopeial friability test and five for the hardness test. The pilot batch with better results was used to manufacture 400 tablets for the dissolution, disintegration, toxicity and pharmacokinetic studies.

### 2.3. Animals

All female rat specimens were purchased from the Universidad Juarez Autónoma de Tabasco (UJAT) bioterium. After arrival at our facilities, they were grouped and kept in acrylic cages under a controlled condition environment (room temperature: $22 \pm 3$ °C; humidity: $55 \pm 15\%$; and a 12–h light/dark cycle). Purified water and pelleted standard chow were delivered ad libitum.

Animal handling and experimentation was performed as dictated by the rules of Association Assessment and Accreditation of Laboratory Animal Care. This experimental protocol was previously approved by the Institutional Animal Care Committee as determined in NOM–062–ZOO–1999.

### 2.4. Acute Oral Toxicity

A single–dose acute oral toxicity assay was performed according to the OECD Test Guideline 423. A total of 300 mg tablets were manufactured for that purpose. Three doses were assessed: 50, 300 and 2000 mg/kg. A group of three rats was used per dose. Toxicity signs and mortality were continuously supervised for 4 h on the first day, two hours the second day and at least one hour per day for an additional 12 days.

### 2.5. Subchronic Oral Toxicity

A repeated dose 28–day oral toxicity test was performed according to the guidelines of OECD Test Guideline 407. Twelve rats were weighted two and four days before the evaluation. Then, they were randomly divided into two groups of six rats each. Pulverized tablets were suspended in 0.9% saline solution for daily oral route administration for 28 consecutive days. The control group received only an equal volume of 0.9% saline solution. The body weight and food and water consumption were registered every day. Five rats of each group were placed in metabolic cages on days 0, 7, 14, 21 and 28 to measure the urine and fecal excretion. Urine samples were analyzed for bacteria, crystals or any other distinct components by microscopic visual tests. Finally, the rats were sacrificed under diethyl ether anesthesia. Blood samples were collected in both EDTA–containing and serum test tubes. The organ (kidney, liver and smooth intestine) weights were measured. Tissue samples were fixed for one week in buffered 10% formalin solution for histological analysis.

### 2.6. Hematological and Biochemical Components Analysis

EDTA tubes were used for whole blood hematology determinations, and serum tubes were used for the aspartate aminotransferase (AST), alanine aminotransferase (ALT), as well as total (BT), indirect (BI) and direct (BD) bilirubin quantifications.

Hematology determination was performed using an automated Sysmex XS–1000i® hematology analyzer (Oak Ridge, TN, USA), while liver function biomarkers were quantified using a Cobas Integra® 400 Plus analyzer (Roche Diagnostics, Rotkreuz, Switzerland) as per the established manufacturer's specifications.

### 2.7. Histopathological Analysis

One–centimeter pieces were processed using the STP 120 tissue processor to obtain Parafilm–embedded tissue samples (Especialidades Médicas Myr, S.L, Tarragona, Spain). These samples were cut into 4–micron thick slices using a HistoCure BIOCUT microtome (Leica Biosystems Division, Leica Microsystems Inc., Chicago, IL, USA). All tissues were

mounted on glass slides (Montval Laboratories SA de CV, Mexico City, Mexico) and stained with hematoxylin–eosin standard dye. Histological samples were visually observed and digitized in a Leica DM300 microscope by v3.0 Leica LAS EZ software. The entire histological analysis process was blinded with control samples as reference slides.

### 2.8. Statistical Analysis

Continuous data are shown as the mean and standard error of the mean (SEM) in all experiments and compared using an unpaired *t*–test. The tissue–mass percentage was calculated as the division of wet organ weight and whole animal body weight (last measure before anesthesia–induced sacrifice). Statistical analysis and plot constructions were carried out in GraphPad Prism software v8.0.2 (San Diego, CA, USA). Statistical significance was established when $p < 0.05$ between an experimental group (different MIX–160 doses) and control group (vehicle solution). Differences are represented with an asterisk symbol.

## 3. Results and Discussion

### 3.1. Acute Oral Toxicity

Overall, we found no animal mortality or toxicity signs after 14 days of visual examination. Biochemical tests showed no modifications of the liver enzymes (ALT/TGP and AST/TGO) and total bilirubin content after treatment (Figure 1).

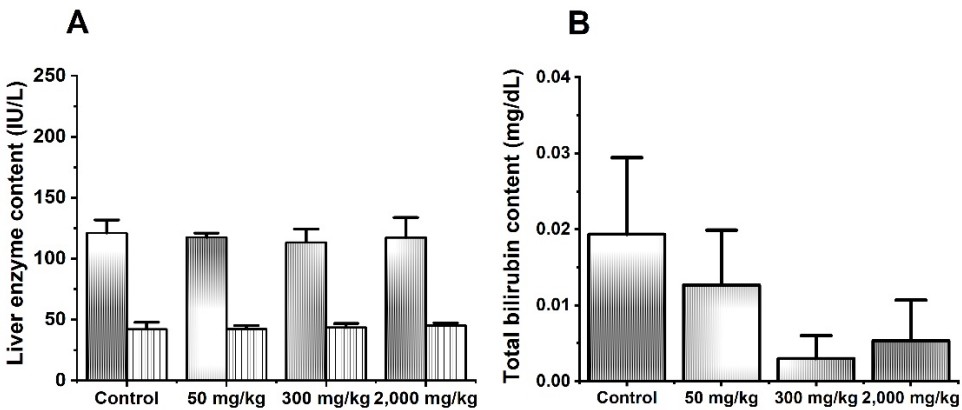

**Figure 1.** Effects of MIX–160 on liver enzyme content (**A**) and bilirubin (**B**) in acute oral toxicity evaluations. ALT is represented in white with gray lines. AST is represented in solid gray.

Previously, Ortiz-Andrade et al. evaluated naringenin and hesperidin (300 and 2000 mg/kg p.o.) in in vivo acute toxicity assay showing a $LD_{50} > 2000$ mg/kg for both flavonoids. Evidence reported by Ortiz-Andrade showed that naringenin and hesperidin were classified as low–risk substances according to OECD guidelines [22]. On the other hand, Yongsheng L. et al. reported that oral administration of hesperidin at 55, 175, 550 and 1750 mg/kg did not produce any signs of toxicity, and all animals survived throughout a 14–day time interval [24].

### 3.2. Subchronic Oral Toxicity

The results showed that all rats survived after a 28–day treatment time interval and showed no apparent signs of toxicity. Body weights were similar between groups throughout the entire study (Figure 2A). Nevertheless, a slight weight loss trend was observed at day 28 in rats treated with MIX–160. However, no statistically significant differences were observed compared to the control group. Kang S. et al., showed that the administration of a flavonoid–rich extract (150 mg/kg for 70 days), including hesperidin, induced weight loss in obese rats. Mechanistic experiments revealed that the lipolysis pathway was activated by phosphorylation of cAMP–dependent protein kinase (PKA) and hormone–sensitive lipase [25]. In this context, further experiments should be performed to probe this possible mechanism exerted by MIX–160.

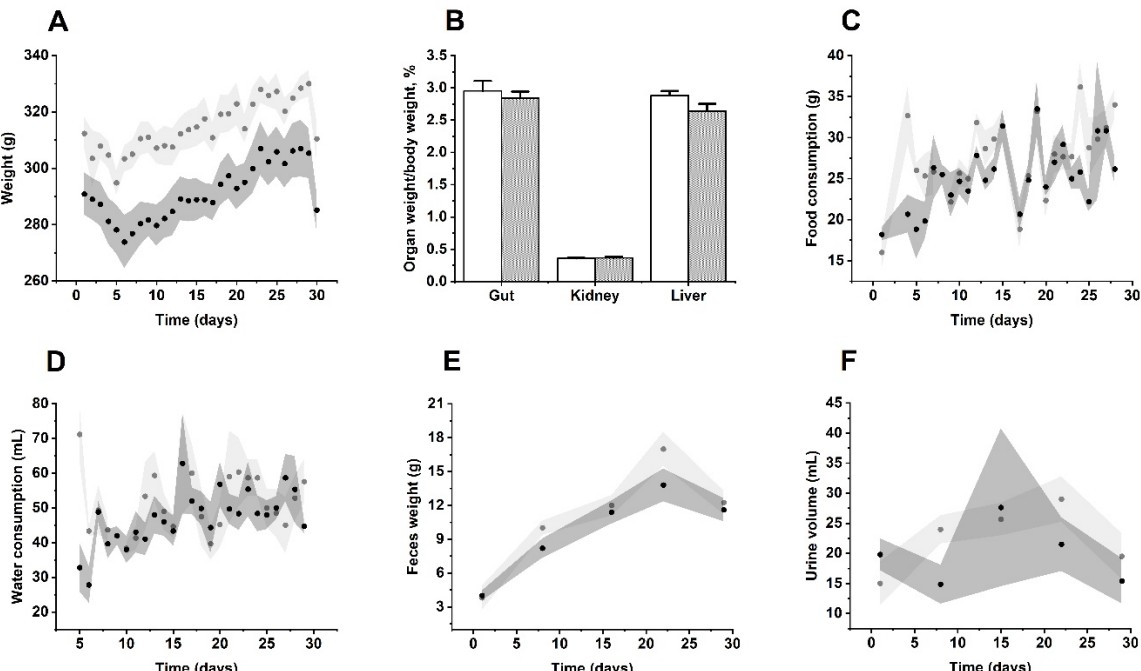

**Figure 2.** Effects of MIX–160 on the (**A**) body weight, (**B**) tissue–mass percentage, (**C**) food consumption, (**D**) water consumption, (**E**) excreted feces and (**F**) urine in repeated dose 28–day oral toxicity evaluations. The control group is represented in light gray or white, and MIX–160 is represented in dark gray.

Taking into account food and water consumption, fecal weight and urine volume, there were no significant changes between MIX–160 and the control group (Figure 2C–F). In addition, microscopic visual examination of the urine did not reveal the presence of bacteria, crystals, or any further alteration. Major tissues and organs (kidney, liver and gut) showed no structural and visual abnormalities between groups (Figure 2B).

### 3.3. Changes of Biochemical and Hematological Parameters by MIX–160

There was no change in the percentage of blood leukocytes between groups (Figure 3A). There were no statistically significant changes in the total bilirubin (Figure 3B), AST and ALT contents (Figure 3C). These findings matched the data reported by Ortiz-Andrade et al. Thus, the total bilirubin, AST and ALT contents were similar in both groups (treated with oral naringenin and hesperidin or control) [22]. Typically, elevated blood liver enzyme content suggest liver damage.

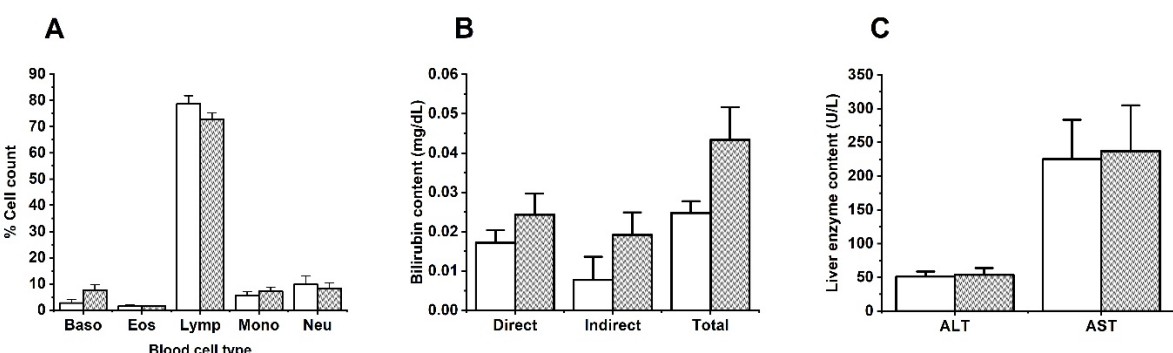

**Figure 3.** Effects of MIX–160 on (**A**) blood cells, (**B**) bilirubin and (**C**) AST and ALT transaminases in repeated dose 28–day oral toxicity evaluations. The control group is represented in white, and MIX–160 is represented in gray.

Chemical substances that are able to revert this process could be potential hepatoprotective agents, as is the case of some flavonoids, such as naringenin, whose administration at doses of 50 and 100 mg/kg in rats with liver damage induced a decrease in transaminases that was accompanied by a reduction in the expression level of proinflammatory cytokines (NF–κβ) [26]. Furthermore, a meta–analysis published by F. Naeini showed that naringenin could be a promising therapeutic agent for the management of non–alcoholic fatty liver disease (NAFLD) and associated complications due to its ability to modify the energy balance, lipid and glucose metabolism, inflammation, and oxidative stress [27].

### 3.4. Histological Findings

The main organ responsible for drug metabolism is the liver, although the kidneys and intestines can also biotransform certain compounds [28]. Most drugs, particularly water–soluble drugs and their metabolites, are eliminated largely by the kidneys in urine [29]. For the reasons mentioned, these organs are vulnerable to developing toxicity and various forms of injury and are an important object of study. The histological samples of liver, intestine and kidney were examined and no treatment–related histopathological changes were recorded in the control or MIX–160 groups (Figure 4).

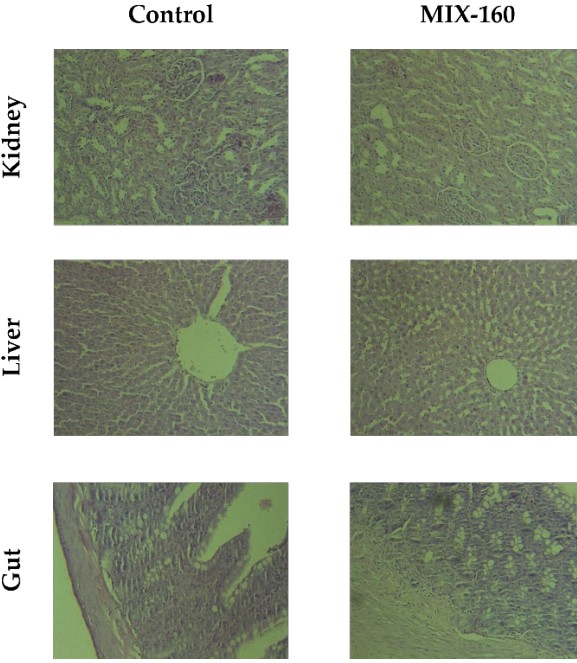

**Figure 4.** Histological evaluation of kidney, liver, and intestine tissues of the control vs. MIX–160 treated groups.

Three different organs were evaluated regarding the histology: the liver, intestines, and kidneys, in different conditions of treatment (acute and sub–chronic exposure) at fixed doses as indicated in the methodology section. As shown in Figure 4, no damage was found in the hepatocyte structure or in the tissue morphology. Some dilated central veins were observed, even in control animals, likely due to the sacrificial method. No signs of chronic or acute cytotoxicity were observed in any of the evaluated conditions. In intestine slides, a slight increase in goblet cells of the intestinal epithelium was observed, especially for acute exposure. To a lesser extent, there was also an increase in these cells in sub–chronic exposure compared to the control. In the kidney preparations, the nephrons, tubules and interstitium were without differences between the control and treated animals.

## 4. Conclusions

Oral single doses (50, 300 and 2000 mg/kg) of MIX–160 did not produce acute oral toxicity in rats. According to these findings, the naringenin–hesperidin mixed product was well tolerated and did not cause either lethality or toxic clinical symptoms when administered at a daily dose of 161 mg/kg for 28 days. The above evidence supports the innocuity and safety profile of MIX–160. Further investigations should be performed to evaluate safety in other rodent and non–rodent species.

**Author Contributions:** Writing—original draft, formal analysis, preparation and creation and/or preparation of the published work, specifically the visualization/data presentation: C.G.C.-S.; conceptualization, formal analysis, funding acquisition, resources, investigation and supervision: R.O.-A.; conceptualization, supervision and writing—review and editing: J.A.A.-L. and M.R.S.-C.; methodology and formal analysis: P.V.-G.; validation and data curation: H.R.-Z. and E.H.-N.; methodology and data curation: E.H.-B. and V.Y.-P.; conceptualization and methodology: A.S.-R.; visualization, editing and critical review: J.C.S.-S. and D.R.-C. All authors have read and agreed to the published version of the manuscript.

**Funding:** This work was supported by CONACYT—Convocatoria para la Atención a Problemas Nacionales 2015: "Estudio preclínico de una forma farmacéutica sólida de citroflavonoides con propiedades hipoglucemiantes e hipotensoras: Caracterización farmacicinética y farmacodinámica" No. 756.

**Institutional Review Board Statement:** The animal study protocol was approved by the Institutional Animal Care Committee as determined in NOM–062–ZOO–1999.

**Informed Consent Statement:** Not applicable.

**Acknowledgments:** All authors would like to thank the "Facultad de Química" and "Facultad de Medicina" from "Universidad Autónoma de Yucatán" and "Facultad de Farmacia" from "Universidad Autónoma del Estado de Morelos" for all the facilities provided in the use of laboratories, equipment, and installations to develop this work. Carla Georgina Cicero Sarmiento thanks CONACYT for her doctoral scholarship (No. 781678) and also thanks the Programa de Maestría y Doctorado en Ciencias Médicas, Odontológicas y de la Salud de la Universidad Nacional Autónoma de México (UNAM). Priscila Vazquez Garcia thanks to CONACYT for granting her the master scholarship (No. 724269).

**Conflicts of Interest:** The authors declare no conflict of interest.

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
