# Peer review of "Preclinical Safety Profile of an Oral Naringenin/Hesperidin Dosage Form by In Vivo Toxicological Tests"

_scipharm, doi:10.3390/scipharm90020028_

Round 1

Reviewer 1 Report

I would suggest a rephrasing of what the study is about, what was planned and performed! It needs to made more clear that you are performing both an acute single dose toxicity study and a subchronic one. 

It would be useful to read a bit more about why the authors chose to study only the listed organs: kidney, liver, and smooth intestine.

What is referenced in lines 218-219 needs rephrasing - makes no sense! ''Chen et al reported that 20-days administration of naringenin (50 and 100 mg/kg) to damage-induced liver rats achieved significantly reduction of transaminases level [23].''

Author Response

Thank you so much for your comments, all of each were very really enriching for our paper, below we attend each of one.

Point 1: I would suggest a rephrasing of what the study is about, what was planned and performed! It needs to make clearer that you are performing both an acute single dose toxicity study and a subchronic one.

Response 1: We change the sentence “We aimed to evaluate oral toxicity of MIX-160 by histological analysis, body weight changes, and biochemical markers quantification. Three different doses (50, 300, 2000 77 mg/kg) and after repeated doses of 161 mg/kg for 28 days“ to “We aimed to evaluate acute and subchronic oral toxicity of MIX-160 by histological analysis, body weight changes, and biochemical markers quantification. Assessments were made at three different doses (50, 300, 2000 mg/kg) as a single dose administration and after repeated doses of 161 mg/kg for 28 days

We also replace the title “Repeated dose 28-day oral toxicity” to “Subchronic oral toxicity” in the Materials and Methods section.

Point 2: It would be useful to read a bit more about why the authors chose to study only the listed organs: kidney, liver, and smooth intestine.

Response 2:

We added the followed paragraph in the 3.4 section: “The main organ responsible for drug metabolism is the liver, although the kidney and intestine can also biotransform certain compounds [28]. Also, most drugs, particularly water-soluble drugs, and their metabolites, are eliminated largely by the kidneys in urine [29]. For the reasons mentioned, these organs are vulnerable to developing toxicity and various forms of injury and are an important object of study. The histological samples of liver, intestine and kidney were examined and observed that no treatment-related histopathological changes were recorded in the control or MIX-160 groups”.

Point 3: What is referenced in lines 218-219 needs rephrasing - makes no sense! ''Chen et al reported that 20-days administration of naringenin (50 and 100 mg/kg) to damage-induced liver rats achieved significantly reduction of transaminases level [23].''

Response 3: Done

Reviewer 2 Report

Dear Authors,

I write you in regard to the manuscript Preclinical safety profile of an oral naringenin/hesperidin dosage form by in vivo toxicological tests.

  • please, add more references for the toxicity of naringenin and hesperidin (lines 71-75)
  • please, consider revising the objectives turning it more attractive to the peers. Last sentence was confusing
  • chemicals were missing in item 2.1
  • please, justify why only female rats were used
  • please, present the definition of NAFLD in line 222
  • overall, this manuscript was very adequately written and could be considered a Communication (short communication) instead of a full article. It is just a suggestion.

Author Response

Thank you so much for your comments, all of each were very really enriching for our paper, below we attend each of one.

Point 1: please, add more references for the toxicity of naringenin and hesperidin (lines 71-75)

Response 1: Done

Point 2: please, consider revising the objectives turning it more attractive to the peers. Last sentence was confusing.

Done, we rephrased last sentence in lines 83-86.

Point 3: chemicals were missing in item 2.1

Response 3: We included the following information:

Microcrystalline cellulose (AVICEL® PH 102; DUPOINT, Dasan, Cd Mx, Mexico) and Agglomerated α-lactose-monohydrate (Tablettose® 70, Meggle AG, WAsserburg am Inn, Germany); Sodium croscaramellose (JRS Pharma, Patterson NY, USA); sodium lauryl sulfate (Cosmopolita Drug Store, Cd. Mx, Mexico); magnesium stearate (Central de drogas, S.A. de CV; Cd Mx, Mexico) excipients used in formulation development were analytical and pharmaceutical grade respectively. 

Point 4: please, justify why only female rats were used

According to the Organization for Economic Cooperation and Development (OECD) Guidelines for the Testing of Chemicals, acute oral toxicity tests should be performed on female rats because they have been shown to have greater sensitivity to male rats in conventional LD50 tests. Also this guidelines supports that when the test is conducted in males adequate justification should be provided.

Reference: OECD, 2001a. Guideline for the Testing of Chemicals: Acute Oral Toxicity—Acute Toxic Class Method (TG 423), adopted 22.03.96: revised method adopted: 17th December 2001. OECD,Paris.

Point 5: please, present the definition of NAFLD in line 222

Done

Reviewer 3 Report

  1. Acute oral toxicity: Three-rats group per dose is too small to testify the results.
  2. Repeated dose 28-day oral toxicity assay: 161 mg/kg showed an improved absorption is not as a reason for choosing dose.
  3. Why use all female rat?

Author Response

Thank you so much for your comments, all of each were very really enriching for our paper, below we attend each of one.

Point 1: Acute oral toxicity: Three-rats group per dose is too small to testify the results.

Response 1: Dear Reviewer 3, thank you so much for your comments. In this manuscript, we adhered to the Organization for Economic Cooperation and Development (OECD) guidelines for testing of chemicals which indicates that for acute toxicity tests should be used three animals of a single sex (normally females).

Reference: OECD, 2001a. Guideline for the Testing of Chemicals: Acute Oral Toxicity—Acute Toxic Class Method (TG 423), adopted 22.03.96: revised method adopted: 17th December 2001. OECD,Paris.

Point 2: Repeated dose 28-day oral toxicity assay: 161 mg/kg showed an improved absorption is not as a reason for choosing dose.

Dear reviewer 3, the dose of 161 mg/kg was chosen because it proved to be the mean effective dose (SD 50) in which we observed an antihyperglycemic effect when administered in diabetic rats, also with this same dose we observed a decrease in aorta blood pressure in our in vitro assays. In addition to previously reported pharmacological effects, based on Sig-ma-Aldrich specifications, naringenin and hesperidin are classified as category 4 acute toxicity substances, with a median lethal dose (LD50) between 300 and <2000 mg/kg. Last reasons, together, justify the dose.

References:

Correa-Polanco I. Determination of the mean effective dose of the hypoglycemic/antihyperglycemic effect caused by a mixture of citroflavonoids. Autonomous University of Yucatán, Mérida, Yucatán, Mexico, 2019.

Sánchez-Rencillas, A.; González-Rivero, N.; Barrea-Canto, V.; Ibarra-Barajas, M.; Estrada-Soto, S.; Ortiz-Andrade, R. Vaso-relaxant and Antihypertensive Activities of Citroflavonoids (Hesperidin/Naringenin Mixture): Potential Prophylactic of Cardiovascular Endothelial Dysfunction. Pharmacogn. Mag. 2019, 15, 84–91. https://doi.org/10.4103/pm.

 Araujo-León, J. A.; Ortiz-Andrade, R.; Vera-Sánchez, R. A.; Oney-Montalvo, J. E.; Coral-Martínez, T. I.; Cantillo Ciau, Z. Development and optimization of a high sensitivity LC-MS/MS method for the determination of hesperidin and naringenin in rat plasma: Pharmacokinetic approach. Molecules, 2020, 25. https://doi.org/10.3390/molecules25184241

Point 3: Why use all female rat?

Response 3: According to the OECD Guidelines for the Testing of Chemicals, acute oral toxicity tests should be performed on female rats because they have been shown to have greater sensitivity to male rats in conventional LD50 tests. Also this guidelines supports that when the test is conducted in males adequate justification should be provided.

Reference: OECD, 2001a. Guideline for the Testing of Chemicals: Acute Oral Toxicity—Acute Toxic Class Method (TG 423), adopted 22.03.96: revised method adopted: 17th December 2001. OECD,Paris.